# Liposomal Vitamin D_3_ as an Anti-aging Agent for the Skin

**DOI:** 10.3390/pharmaceutics11070311

**Published:** 2019-07-03

**Authors:** Ye Bi, Hongxi Xia, Lianlian Li, Robert J. Lee, Jing Xie, Zongyu Liu, Zhidong Qiu, Lesheng Teng

**Affiliations:** 1Key Laboratory of Effective Components of Traditional Chinese Medicine, Changchun University of Chinese Medicine, Changchun 130117, Jilin, China; 2Practice Training Center, Changchun University of Chinese Medicine, Changchun 130117, Jilin, China; 3School of Life Sciences, Jilin University, Changchun 130117, Jilin, China; 4Division of Pharmaceutics, College of Pharmacy, The Ohio State University, Columbus, OH 43210, USA; 5Department of Clinical Medicine, Norman Bethune Health Science Center of Jilin University, Changchun 130117, Jilin, China; 6Department of Pharmacy, Changchun University of Chinese Medicine, Changchun 130117, Jilin, China

**Keywords:** vitamin D_3_, liposomes, prevent photoaging, stability

## Abstract

Vitamin D_3_ is an effective skin protective substance to prevent photoaging. Liposomes were used as a carrier to deliver vitamin D_3_ to improve the stability and to enhance the treatment effect of vitamin D_3_. The stability of vitamin D_3_ liposomes, average cumulative penetration, and retention of vitamin D_3_ in the skin were then evaluated and compared with free vitamin D_3_. Finally, the treatment effect of vitamin D_3_ liposomes in a rat photoaging model was appraised and Haematoxylin-Eosin (H&E) staining was used to assess the histology changes of the skin after vitamin D_3_ liposome treatment. The results indicated that liposomes could significantly improve the stability of vitamin D_3_. The average skin retention of vitamin D_3_ liposomes was 1.65 times that of the vitamin D_3_ solution. Vitamin D_3_ liposomes could repair the surface morphology of skin in the photoaging model and promote the production of new collagen fibers. Vitamin D_3_ liposomes as a potential skin care agent could significantly improve skin appearance and repair damage in the histology of photoaging.

## 1. Introduction

Skin aging is a progressive and comprehensive process driven by the environment and time. It is associated with aesthetic and functional changes, such as an increase in wrinkles and cutis laxa. Skin aging is a slow and irreversible process. Photoaging occurs due to perennial exposure to ultraviolet light (UV), which causes skin damage [1]. Long-term UV irradiation induces reactive oxygen species (ROS) generation. Oxygen free radicals destroy DNA, the extracellular matrix and membrane lipids of skin cells and cause the development of wrinkles [2]. In addition, cellular damage (e.g., DNA damage due to ROS generation) induces skin cells to overexpress matrix metalloproteinases, which destroy the skin matrix tissue [3,4]. Histological studies have shown that major damage occurs in skin connective tissue upon UV irradiation [5].

Vitamin D_3_ (VD_3_) is known for its capacity to maintain bone health. VD_3_ has an impact on the skin, although it is produced in the skin. Recent studies have shown that VD_3_ could repair skin cell damage caused by UV. The main damage by UV is to the DNA by the induction of cyclobutane pyrimidine dimers (CPDs) [6,7]. VD_3_ significantly reduces the induction of CPDs through increasing two nucleotide excision repair by inducing expression of damage-specific DNA binding protein 2 and xeroderma pigmentosum complementation group C [8]. In addition, VD_3_ increases the expression of p53 protein, which enhances DNA repair by blocking the progression of the cell cycle [9,10]. UV also induces an increased level of ROS and NO in skin cells, which increases oxidative damages, such as DNA strand breaks, oxidation of purine or pyrimidine, and lipid peroxidation. VD_3_ could reduce ROS and NO generation induced by UV. Furthermore, it protects keratinocytes against UV damage [10,11,12,13]. The idea of sunburn and VD_3_ preventing photoaging may seem counter-intuitive, however, skin exposed to UV is the major mechanism to obtain VD_3_ for humans. But the VD_3_ photosynthesis, erythema, and CPD formation are all in the same spectral range of UVB. Therefore, the photosynthesis of VD_3_ can not be dissociated from photodamage. Although VD_3_ could be transformed under UV, it only occurs in the first few minutes upon exposure to UV, and longer exposure adds nothing to VD_3_ stores despite increasing DNA damage in a linear fashion. So, naturally occurring VD_3_ under UV is insufficient to prevent photoaging. Additional VD_3_ needs to be supplied for skin protection [14]. 

Photostability and effectiveness are two key parameters of sunscreen products [2]. Although VD_3_ effectively prevents skin photoaging, VD_3_ is sensitive to air, light and high temperature, which is not conducive to storage, transportation and use [2].VD_3_ also rapidly degrades in the traditional carrier (water/ethanol) [15]. Liposomes are commonly used as a carrier to protect drugs from degradation. Liposomes are also used as a protective carrier in cosmetics to enhance the stability of active substances [16]. The poor solubility of VD_3_ impedes uniform distribution in the skin. Liposomes also could improve solubility and control drug release [17].

In this study, we optimized the liposomal formulation of VD_3_ to improve encapsulation efficiency (EE%) and to enhance transdermal absorption and stability of VD_3_.

## 2. Materials and Methods

### 2.1. Materials

Vitamin D_3_ (crystalline, purity > 99%) was purchased from Shanghai Yuanye Bio-Technology Co., Ltd. (Shanghai, China). Egg phosphatidylcholine (egg PC) was purchased from Jinban Pharmaceutical Co., Ltd. (Shanghai, China). Cholesterol (Chol) was obtained from Beijing Dingguo Changsheng Biotechnology Co., Ltd. (Beijing, China). Other reagents were all of the analytical grade. Aloe-gel was purchased from PERFECT Co., Ltd. (Guangdong, China).

### 2.2. VD_3_ Liposomes Preparation

Particle size and EE% were used as assessment criteria to evaluate two common liposomes preparation methods of VD_3_ (ethanol injection, film dispersion-homogenization method). A mixture of 30 mg of phospholipids, 10 mg of Chol and 10 mg of VD_3_ was dissolved in 5 mL ethanol as the lipid phase.

Ethanol injection method: The lipid solution was fast injected into 200 mL phosphate buffer solution (pH 6.8, PBS) under vortex mixing for 30 s. Then ethanol was removed by rotary evaporation and the liposomes were condensed to 100 mL under 35 °C. Finally, the liposomes of VD_3_ were sonicated for 5 min with an ice-bath.

Film dispersion-homogenization method: The lipid solution was dried by rotary evaporation at 35 °C to form a film. Then, 200 mL of PBS was used to hydrate lipid and liposomes were also condensed to 100 mL. Finally, liposomes were sonicated for 5 min in an ice-bath and were subjected to high-pressure homogenization (150 MPa) three times. 

### 2.3. Encapsulation Efficiency

The EE% of VD_3_ was measured by filter method. Free VD_3_ was separated from liposomes by ultrafiltration centrifuge tube. Briefly, methanol was used to release and determine total VD_3_ (W_total_) in liposomes. On the other hand, an ultrafiltration centrifuge filter (cut off MW of 10 kDa) was used to separate free VD_3_ (W_free_) from liposomes. The concentrations of W_total_ and W_free_ were then measured by high-performance liquid chromatography (HPLC) method.

The concentration of VD_3_ was determined by HPLC at an absorption wavelength of 204 nm with 1 mL/min methanol as the mobile phase using a C18 reversed-phase column (4.6 × 250 mm, 5 μm).

EE% = (1 − W_free_/W_total_) × 100%

### 2.4. Characterization of VD_3_ Liposomes

The particle size of VD_3_ liposomes was measured by Zetasizer Nano ZS 90 from Malvern Instruments, Ltd. (Malvern, U.K.). The size was shown by intensity-weighted distribution in Gaussian mode. A JEOL, JSM-6700F field emission scanning electron microscope (SEM) (Tokyo, Japan) was used to evaluate the surface morphology of VD_3_ liposomes at 3 kV accelerating voltage. VD_3_ liposomes were directly dropped on a clean silicon slice to dry naturally at room temperature overnight. 

### 2.5. Optimization of Liposomes Formulation

On basis of single-factor experiments, we chose three interrelated factors *X1* (the amount of egg PC), *X2* (the mass ratio of egg PC to Chol), and *X3* (the mass ratio of lipid to the drug) to optimize the liposomes formulation using drug EE% as an evaluation indicator. We designed 17 experiments of three-factor, three-coded level using Box-Behnken design (BBD) to obtain the optimal VD_3_ liposomes formulation. Liposomes were prepared by the ethanol injection method. The influence of factors on EE% were shown by response surface method. The factor-level table was listed in Table 1. The optimal formulation of VD_3_ liposomes was calculated by fitting the equation between EE% and three factors. The optimal formulation of VD_3_ liposomes was verified three times in repetitive experiments.

### 2.6. Stability of VD_3_ in Liposomes

The stability of VD_3_ liposomes was evaluated in different conditions, including variations in time elapsed, variations in temperature, light exposure and after freezing and thawing. EE% and drug-content were used to estimate the stability of VD_3_ liposomes. We prepared 100 μg/mL of VD_3_ liposomes to compare with VD_3_ solution.

Samples were placed under 4, 25 and 40 °C in darkness and evaluated after storing for 1, 3, 5, 7 or 9 days. To study the effect of light on the stability of liposomes, 4500 Lx light was used to irradiate liposomes for 10 days at 25 °C. At each sampling time (1, 3, 5, 7 or 9 days), we analyzed the evaluation index of VD_3_ liposomes to study stability. Finally, VD_3_ liposomes were measured after freezing at −70 °C and thawing at 25 °C.

### 2.7. Transdermal Absorption of VD_3_ Liposomes

Wistar rats were fed according to the cleanliness class specifications, and 24 h free diet. Before the transdermal absorption experiment, a rat was euthanized with sodium pentobarbital and sacrificed. A piece of rat back skin was harvested without hair and after removing skin moisture with filter paper, the rats using protocol and relevant experimental procedures was approved by the Institutional Animal Care and Use Committee of Jilin University at January, 2017 (Authorization number: 201701005). The rat skin was fitted between the accept pool and supply pool of Franz diffusion pool. The corneous layer-oriented supply pool and diffusion area was 2.54 cm^2^. One ml of VD_3_ liposomes or VD_3_ solution (1 mg/mL) was evenly daubed onto the skin, while the supply pool was covered by plastic wrap. The accept pool was filled with saline solution containing 30% (*v/v*) ethanol under stirring (400 rpm) and maintained at a constant temperature of 37 °C. The device was placed in darkness. At sampling time of 0.5, 1, 2, and 4 h, we collected 1 mL receiving liquid to test the concentration of VD_3_, then replaced equivalent fresh liquid to the accept pool.

The residual VD_3_ on the skin surface was cleaned, the skin was then cut into pieces. Pieces of skin were homogenized with PBS by a high-speed homogenizer. VD_3_ within the skin homogenate solution was extracted by liquid-liquid extraction. Briefly, 200 ul skin homogenate solution was added into 1 mL methanol with vortex for 30s. And then 1 mL of hexane was added to extract VD_3_ under oscillation mixture for 2 min [18]. The mixture was centrifuged at 2000 rpm for 5 min to separate supernatant fluid. The supernatant fluid containing VD_3_ was dried by nitrogen at 4 °C. Finally, the sample was dissolved in 200 μL of HPLC initial mobile phase. The skin retention of VD_3_ was measured by the HPLC method.
$$Q=\left[\overset{n-1}{\underset{i=1}{\sum }}\mathrm{Ci}*V+\;\mathrm{Cn}*V0\right]/S$$

*Q*: Average cumulative penetration of VD_3_ (μg/cm^2^)

Cn: Concentration of VD_3_ at the nth hour.

Ci: Concentration of VD_3_ when the ith.

V0: Volume of the accept pool (6.50 cm^2^)

V: Volume of the sampling

S: Diffusion area

### 2.8. VD_3_ Liposomes for Preventing Photoaging

Animal experiments were approved by the Institutional Animal Care and Use Committee of Jilin University at January, 2017 (Authorization number: 201701005). Wistar rats were fed according to the cleanliness class specifications.

Wistar rats were divided into four groups, each group contained eight rats with half female and half male. Rats were normally fed for 7 days with 24 h free diet to adapt to the environment before the experiment.

Back hair of rats in four groups was cleanly shaved in an area of about 3 × 5 cm. Ultraviolet ray was used to illuminate the exposed skin of rats to build a photoaging model. The exposed skin of rats was continuously irradiated for 1 month by 30 W of UVA and UVB ultraviolet lamp placed 30 cm above the cage, each irradiation time was 2 h every two days. 

The four photoaging model groups were respectively administrated PBS, 1 mL of 100 μg/mL VD_3_ solution, VD_3_ liposomes or 1 mL of aloe-gel twice per day. The rats were continuously treated for 1 month. Apparent properties of the rats’ skin were quantitatively analyzed after treatment using the skin color and scale. And then the four groups of rats were anesthetized and sacrificed to harvest the exposed target skin, the skin was fixed by 4% paraformaldehyde, embedded in paraffin. Finally, the skin tissues were stained by the H&E method to analyze the histology of epidermis, dermis, collagen fibers and elastic fibers.

### 2.9. Statistical Analysis

A t-test was used to analyze the statistical significance of the data. P values < 0.05 and < 0.01 indicate significant difference and extremely significant difference respectively. The results are shown as mean ± SD.

## 3. Result and Discussion

### 3.1. Preparation of VD_3_ Liposomes

VD_3_ was an effective skin care product to prevent photoaging as a supplement in cosmetics. But the stability of VD_3_ was an important limiting factor in practical application. VD_3_ is a highly liposoluble vitamin and can be embedded into the lipid layer leading to high encapsulation efficiency. We used liposomes as a VD_3_ delivery carrier to improve stability and treatment efficacy.

We compared two common VD_3_ liposome preparation method (ethanol injection, and film dispersion-homogenization) to investigate the particle size and EE%. The results are shown in Table 2. The film dispersion-homogenizing method resulted in multivesicular liposomes, which had a relatively larger size and polydispersity index (PDI). Ethanol injection method obtained uniform size liposomes (92.5 nm, PDI = 0.085) with higher EE% for VD_3_. Finally, we chose the ethanol injection method to prepare VD_3_ liposomes in the subsequent experiments.

### 3.2. Optimization of VD_3_ Liposomes Formulation

On basis of single factor experiments, we used three interrelated factors *X1* (the amount of egg PC), *X2* (the mass ratio of egg PC to Chol), and *X3* (the mass ratio of lipid to the drug) as dependent variates. 17 experiments were designed by Box-Behnken design to optimize the optimal formulation using EE% as a response value. Results were listed in Table 3 and Table 4 along with the statistical results of regression analysis.

The EE% ranged from 59.32% to 84.11% in the 17 experiments. We built a fitting equation between factors and EE% by regression analysis of the results. The fitting equation between EE% and three factors was: EE% = 83.48 + 0.58*X1*+ 7.35*X2*+ 1.95*X3* − 0.60*X1X2* − 4.47*X1X*3 + 0.35*X2X3* − 6.56*X1*^2^ − 4.18*X2*^2^ − 11.17*X3*^2^.

The P value of the mathematical model was <0.0001, R-Squared = 99.5%, which implied the model was significant. The “Lack of Fit P-value” of 0.0899 implied that the model misfits with a small probability. The model was accurate and reliable, it could be used in EE% prediction. The parameters “*X2, X3, X1X*3, *X1*^2^, *X2*^2^, *X3*^2^” were significant model terms in the model, it meant that the factors and EE% were not simple linear relations. The interaction of the two factors was signified in Figure 1. There was an obvious angle between elliptical and the coordinate axis in Figure 1A, it was shown that with the two factors (*X1X3*) existed the most significant interaction. We calculated the optimal formulation was 24.55 mg of egg PC, 5.16 mg of Chol and 11.34 mg of VD_3_, the predicted EE% was 86.82%. The predicted result was verified three times; the mean EE% of VD_3_ was 86.77% and the experimental value fit with the predicted value. Optimized liposomes were characterized in Figure 2. In Figure 2A, the mean size of VD_3_ liposomes was 99.7 nm, the zeta potential was −19.1 mV. The solubility of VD_3_ was significantly improved after envelopment in liposomes. VD_3_ liposomes exhibited as a transparent solution when the concentration was at 1 mg/mL and 100 μg/mL (Figure 2B). In Figure 2C, the surface morphology of VD_3_ liposomes were uniform spheres.

### 3.3. The Stability of VD_3_ Liposomes

Stability of VD_3_ is a major matter in the commercial application. VD_3_ could be easily degraded by various environmental factors (time, temperature and light). We used liposomes to improve the stability of VD_3_. Results are shown in Table 5 and Table 6. The VD_3_ solution was degraded by more than 20% in ethanol/water solvent at 4, 25, and 40 °C. Especially at 40 °C the content of VD_3_ reduced by more than 50% after storing 9 days. However, when the VD_3_ was encapsulated into liposomes, the EE% and content did not significantly decrease at 4 °C and 25 °C. There was about an 8.4% reduction of EE% and 9.3% decline of content when the temperature ascended to 40 °C. Particle size and PDI of VD_3_ liposomes had a tendency to increase during storage.

Stability of VD_3_ liposomes in light suggested that liposomes could protect VD_3_ from photo-degradation. As shown in Table 6, the VD_3_ liposomes shown more stability with light irradiation at 4500 Lx compared to VD_3_ solution. The EE% of liposomes did not have a significant decrease after light irradiation of 9 days. The content of VD_3_ in liposomes decreased by about 5.32% compared to 48.2% of VD_3_ solution after light irradiation for 9 days. In addition, EE% and content of VD_3_ liposomes were only reduced by 0.51% and 1.18% after freezing and thawing.

### 3.4. Transdermal Absorption of VD_3_

We respectively measured the average cumulative penetration and retention of VD_3_ in the skin, the results are shown in Table 7. The total skin uptake of VD_3_ solution and VD_3_ liposomes were all about 50%. Average cumulative penetration of VD_3_ solution was about 5.2 times that of VD_3_ liposomes. However, average retention of VD_3_ liposomes was 1.65 times that of the VD_3_ solution. This indicates that the VD_3_ solution easily penetrated skin tissue. VD_3_ liposomes tended to be absorbed and stranded by the skin, which might be due to the particle size of VD_3_ liposomes. It might improve the effect of VD_3_ on photoaging. 

### 3.5. VD_3_ Liposomes to Prevent Photoaging

Skin photoaging referred to long-term skin damage caused by ultraviolet radiation. VD_3_ could effectively improve the situation. VD_3_ liposomes were prepared to enhance the skin retention of VD_3_ and improve the curative effect. Firstly, a photoaging model of rats was established through UV radiation. And then PBS, VD_3_ solution, VD_3_ liposomes or aloe-gel were used to treat rats, respectively. The result was shown in Figure 3. Skin premature aging, relaxation, roughness and a series of changes of morphology could be found in Figure 3A. In the treatment groups (Figure 3B–D), it could be found that skin redness, loss of luster, roughness, abnormal thickening problems were variously improved after administration of VD_3_ solution, VD_3_ liposomes or aloe-gel for 1 month. The VD_3_ liposomes had the best treatment effect based on the improvement in skin external appearance. The skin color and scale were used as evaluation indexes of VD_3_ liposome treatment effect. The skin color of rats after treatment was shown in Figure 4. The skin color of rats after treatment was significantly lower than the PBS group, except male rats in the aloe-gel group. Skin appearance of rats was counted by scale and results were listed in Table 8. Male rats in the VD_3_ liposomes group and female in VD_3_ solution, VD_3_ liposomes or aloe-gel group all received an acceptable therapeutic effect. VD_3_ liposomes had a protective effect on photoaging.

The tissue sections of treated rats were stained to further study histology changes of the skin by H&E staining. The results are shown in Figure 5. 

The rat skin of the control group was with complete structure and moderate thickness. The dermis had a compact structure and collagen fibers were evenly distributed along the epidermis (Figure 5A). 

Photoaging model group exhibited uneven epidermis thickening, the dermis structure was disrupted with thinning, collagen fibers swelling, homogenizing, and follicle hyperplasia (Figure 5B).

The VD_3_ solution could slightly improve the structural integrity of photoaging skin, and promote the production of new collagen fibers, however the structure of the dermis was relatively loose (Figure 5C).

VD_3_ liposomes had the best treatment effect on photoaging. The VD_3_ liposomes fundamentally restored the structure of photoaging skin. The boundary between the epidermis and dermis layer was very clear. The newborn collagen fibers and glands were orderly lined with compact structures (Figure 5D). The treatment effect of VD_3_ liposomes was similar to the positive control of aloe-gel (Figure 5E).

The treatment of PBS resulted in no improvement for photoaging skin (Figure 4F). The treatment effect of VD_3_ solution, VD_3_ liposomes and aloe-gel had no obvious selectivity for gender.

## 4. Conclusions

Excessive exposure to sunlight causes skin photoaging. The severity of this complication mainly depends on genetic factors, skin type, and the area of skin exposed to the sun. The skin produces high-level ROS upon UV radiation, which damages skin cells. In addition, UV radiation might directly damage DNA and cause skin cancer.

VD_3_ was found to have a good protective effect from UV radiation and photoaging. Of course, there are still some remaining issues to be overcome, such as stability and skin uptake of VD_3_. In 1987, Christian Dior launched the first liposomes cosmetics “Capture”. Since then, this class of cosmetics has received positive attention. About 300 liposomal cosmetics have been commercialized so far. In this study, we prepared VD_3_ liposomes to prevent photoaging. The VD_3_ liposomes had high drug EE% and uniform particle size. Liposomes raised the retention amount of VD_3_ in the skin by transdermal absorption compared to the VD_3_ solution. Finally, VD_3_ liposomes could significantly improve skin appearance and repair damage in the histology of photoaging. This suggests that VD_3_ liposomes warrant further research as a protective agent against photoaging in the skin care field.

## Figures and Tables

**Figure 1 pharmaceutics-11-00311-f001:**
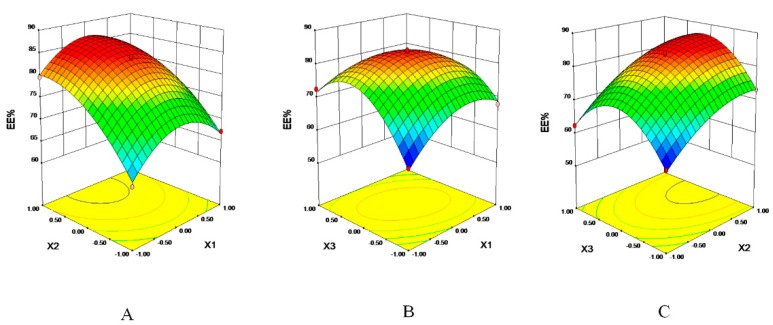
Response surfaces and contours. (**A**) Interaction of *X1* and *X2**,* (**B**) Interaction of *X1* and *X3*, (**C**) Interaction of *X2* and *X3.*

**Figure 2 pharmaceutics-11-00311-f002:**
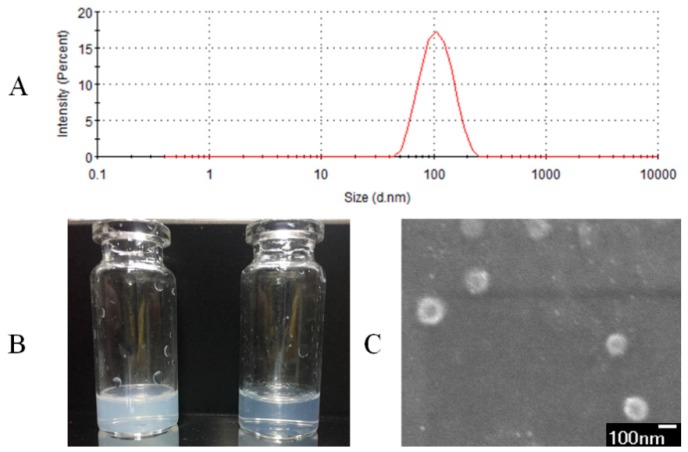
Characterization of vitamin D_3_ (VD_3_) liposomes. (**A**) The particle size of VD_3_ liposomes. (**B**) Morphological appearance of VD_3_ liposomes at 1 mg/mL and 100 μg/mL. (**C**) Scanning electron microscope (SEM) of VD_3_ liposomes at secondary electron imaging mode (×30,000).

**Figure 3 pharmaceutics-11-00311-f003:**
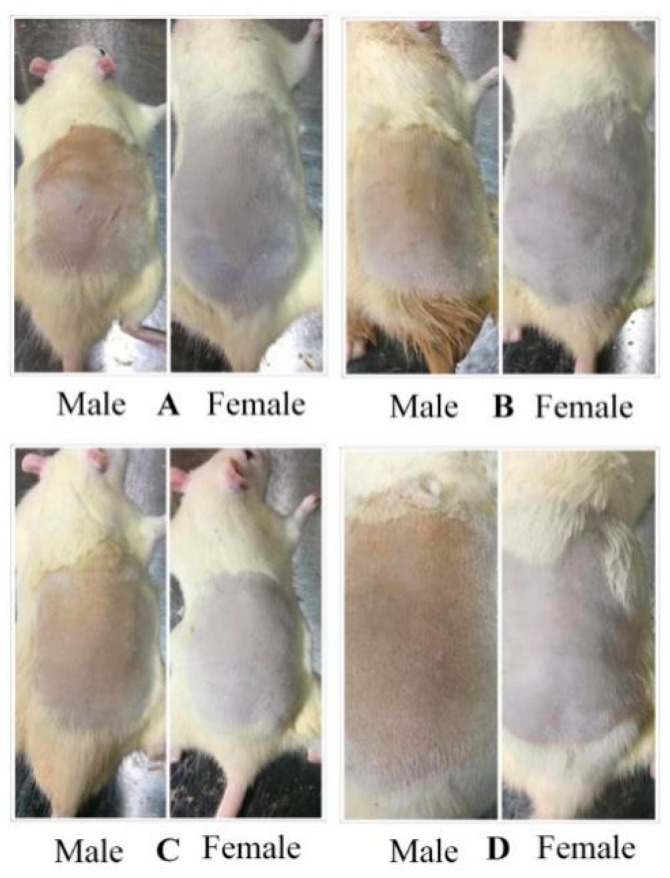
The skin condition of rats after administration VD_3_ solution, VD_3_ liposomes or aloe-gel for one month. (**A**) Treatment with PBS as a control group. (**B**) Treatment with VD_3_ solution. (**C**) Treatment with VD_3_ liposomes. (**D**) Treatment with aloe-gel as a positive control group.

**Figure 4 pharmaceutics-11-00311-f004:**
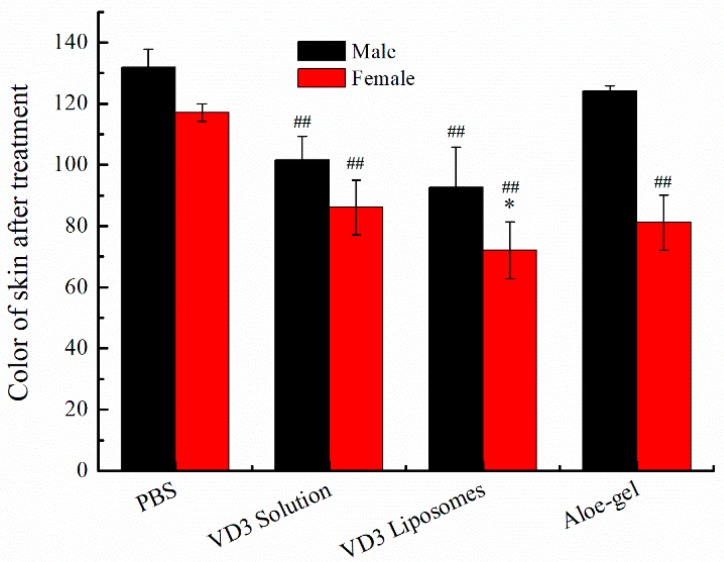
Color of rats’ skin after treatment. * *p* < 0.05, VD_3_ liposomes vs. VD_3_ solution for female rats; ^##^
*p* < 0.01, VD_3_ solution, VD_3_ liposomes or aloe-gel vs. PBS for male and female rats.

**Figure 5 pharmaceutics-11-00311-f005:**
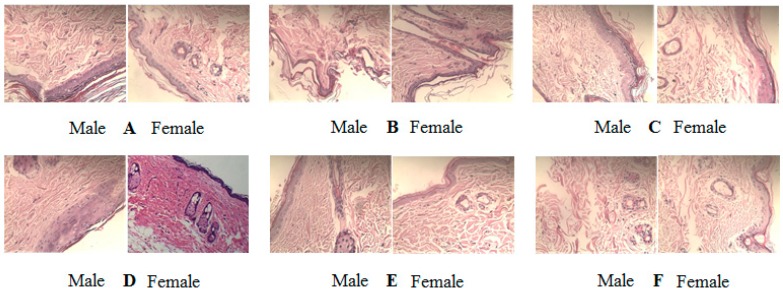
The histology of skin by H&E staining of the skin tissue section. (**A**) Control, normal skin (**B**) Photoaging model, (**C**) Treatment with VD_3_ solution, (**D**) Treatment with VD_3_ liposomes, (**E**) Treatment with aloe-gel as positive control (**F**) Treatment with PBS. H&E 200x.

**Table 1 pharmaceutics-11-00311-t001:** Levels of factors used in Box-Behnken design (BBD).

Factors	Range and Level		
	−1	0	1
*X1* (the amount of egg phosphatidylcholine (PC))	10 mg	25 mg	40 mg
*X2* (the mass ratio of egg PC-to-cholesterol (Chol))	1:1	3:1	5:1
*X3* (the mass ratio of lipid-to-drug)	1:2	2:1	3.5:1

**Table 2 pharmaceutics-11-00311-t002:** Effect of different VD_3_ liposomes preparation methods on the encapsulation efficiency and size distribution (*n* = 3).

Evaluation Index	Film Dispersion-Homogenizing	Ethanol Injection
Encapsulation efficiency (EE%)	62.2 ± 0.9	80.3 ± 0.4
Particle size (nm)	169.4 ± 2.0	92.5 ± 1.8

**Table 3 pharmaceutics-11-00311-t003:** The matrix of Box-Behnken design and the responding experimental results.

Std	Run	Levels of Independent Fctors			Response EE
		*X1*	*X2*	*X3*	(%)
12	1	25 mg	5:1	3.5:1	77.35
13	2	25 mg	3:1	2:1	83.79
5	3	10 mg	3:1	1:2	59.32
9	4	25 mg	1:1	1:2	59.61
8	5	40 mg	3:1	3.5:1	63.24
4	6	40 mg	5:1	2:1	81.04
17	7	25 mg	3:1	2:1	83.35
14	8	25 mg	3:1	2:1	83.68
15	9	25 mg	3:1	2:1	84.11
10	10	25 mg	5:1	1:2	73.15
16	11	25 mg	3:1	2:1	82.45
11	12	25 mg	1:1	3.5:1	62.39
2	13	40 mg	1:1	2:1	67.11
6	14	40 mg	3:1	1:2	67.87
3	15	10 mg	5:1	2:1	79.55
7	16	10 mg	3:1	3.5:1	72.56
1	17	10 mg	1:1	2:1	63.22

**Table 4 pharmaceutics-11-00311-t004:** Statistical results of regression analysis.

Source	Sum of	df	Mean	F Value	P Value
	Squares		Squares		Prob > F
Model	1400.43	9	155.60	154.33	<0.0001
*X1*	2.66	1	2.66	2.63	0.1486
*X2*	431.59	1	431.59	428.05	<0.0001
*X3*	30.38	1	30.38	30.13	0.0009
*X1X2*	1.44	1	1.44	1.43	0.2710
*X1X3*	79.83	1	79.83	79.18	<0.0001
*X2X3*	0.50	1	0.50	0.50	0.5024
*X1^2^*	181.29	1	181.29	179.8	<0.0001
*X2^2^*	73.72	1	73.72	73.11	<0.0001
*X3^2^*	525.04	1	525.04	520.73	<0.0001
Residual	7.06	7	1.01	-	-
Lack of Fit	5.45	3	1.82	4.51	0.0899
Pure Error	1.61	4	0.40	-	-
Cor Total	1407.49	16	-	-	-

**Table 5 pharmaceutics-11-00311-t005:** The stability of VD_3_ liposomes in different time and temperature. Polydispersity index (PDI).

**4 °C**	**VD_3_ Solution**	**VD_3_ Liposomes**		
**Day**	**Content (μg/mL)**	**Content (μg/mL)**	**EE%**	**Size (nm)/PDI**
1	98.74 ± 1.69	96.56 ± 2.35	87.24 ± 1.07	107.7/0.11
3	93.56 ± 1.19	96.37 ± 1.03	87.11 ± 1.40	115.4/0.18
5	87.34 ± 1.01	95.74 ± 0.63	86.88 ± 1.65	126.2/0.15
7	82.11 ± 1.09	95.15 ± 1.75	86.58 ± 1.39	133.0/0.25
9	78.40 ± 1.66	94.89 ± 1.75	86.41 ± 1.37	150.9/0.33
**25 °C**	**VD_3_ Solution**	**VD_3_ Liposomes**		
**Day**	**Content (μg/mL)**	**Content (μg/mL)**	**EE%**	**Size (nm)/PDI**
1	97.27 ± 1.61	95.24 ± 0.34	86.69 ± 1.58	109.6/0.15
3	92.08 ± 2.29	95.13 ± 1.74	86.58 ± 0.72	104.7/0.17
5	85.46 ± 1.98	94.88 ± 0.61	85.17 ± 0.40	113.6/0.17
7	78.31 ± 2.53	94.60 ± 0.86	85.24 ± 1.33	120.9/0.21
9	73.36 ± 1.31	93.98 ± 1.45	85.09 ± 0.32	135.9/0.25
**4** **0 °C**	**VD_3_ Solution**	**VD_3_ Liposomes**		
**Day**	**Content (μg/mL)**	**Content (μg/mL)**	**EE%**	**Size (nm)/PDI**
1	82.40 ± 2.72	91.93 ± 0.56	82.16 ± 0.97	111.2/0.17
3	73.44 ± 2.47	91.09 ± 0.59	81.98 ± 1.98	125.1/0.15
5	66.97 ± 1.48	90.81 ± 0.79	81.22 ± 2.65	115.7/0.19
7	59.38 ± 0.75	90.38 ± 1.73	80.74 ± 2.39	127.3/0.24
9	51.17 ± 0.61	89.57 ± 0.39	79.39 ± 0.55	129.8/0.39

**Table 6 pharmaceutics-11-00311-t006:** The stability of VD_3_ liposomes in 25 °C and 4500 Lx light.

Day	VD_3_ Solution	VD_3_ Liposomes		
	Content (μg/mL)	Content (μg/mL)	EE%	Size (nm)/PDI
1	91.35 ± 0.50	96.07 ± 1.05	86.43 ± 2.76	103.1/0.17
3	83.34 ± 1.21	95.73 ± 0.19	86.09 ± 0.46	108.6/0.19
5	72.90 ± 1.02	95.21 ± 1.78	85.33 ± 1.39	132.6/0.16
7	63.11 ± 2.56	95.38 ± 0.89	85.47 ± 0.26	130.9/0.23
9	50.48 ± 2.15	93.58 ± 2.52	83.69 ± 0.62	137.3/0.27

**Table 7 pharmaceutics-11-00311-t007:** Average cumulative penetration and retention of VD_3_ in the skin.

Time (h)	Average Cumulative Penetration		Run	Average Retention in Skin	
	VD_3_ Solution (μg/cm^2^)	VD_3_ Liposomes (μg/cm^2^)		VD_3_ Solution (μg/cm^2^)	VD_3_ Liposomes (μg/cm^2^)
0.5	2.97	1.14	1	112.97	184.87
1	26.92	4.13	2	116.92	178.29
2	61.56	8.55	3	111.56	183.22
4	80.29	15.48	-	-	-

**Table 8 pharmaceutics-11-00311-t008:** Assessment of photographs of photoaging rats.

Gender	Assessment	Phosphate Buffer Solution (PBS)	VD_3_ Solution	VD_3_ Liposomes	Aloe-Gel
Male	Wrinkles ^a^	Na	Na	Na	Na
	Mottles ^a^	Na	1	Na	1
	Photodamage ^a^	3	2	Na	Na
	Roughness ^a^	4	3	3	4
	Overall appearance ^b^	+++++++	++++++	+++	+++++
Female	Wrinkles ^a^	Na	Na	Na	Na
	Mottles ^a^	Na	1	Na	1
	Photodamage ^a^	1	Na	Na	Na
	Roughness ^a^	3	2	1	1
	Overall appearance ^b^	++++	+++	+	++

^a^: Based on a five-point scale: 1, excellent; 5, poor.Na: not available. ^b^: + and ++, satisfactory; +++ and ++++, acceptable; from +++++ to +++++++, poor.

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
