# Peer review of "Liposomal Vitamin D3 as an Anti-aging Agent for the Skin"

_pharmaceutics, 2019, doi:10.3390/pharmaceutics11070311_

Round 1

Reviewer 1 Report

The manuscript by Bi et al describe the preparation of liposomes containing VitD3 as potential skin protection for sun photoaging. In general, the manuscript is well written and the results are clearly described and discussed.

However, there are still some issues that should be addressed before the manuscript can be accepted for publication.

Page 4, line 170 Table 2. It is not clear to which liposome preparation belong the data of Table 2. The authors report under M&M three different compositions. Please clarify.

Page 5, line 179, Table 3. The amount of eggPC reported under column X1 should be 10, 25 and 40 mg not 0, -1 and 1.

Which is the rationale behind the fitting analysis performed in order to correlate the EE% with the different proportions of lipids?

Why did the authors never prepare liposomes without cholesterol? Or in which the amount of VitD3 incorporated is equivalent to the amount of cholesterol reduced.

Page 11, lines 286-287. Although the results are promising it is not correct to extrapolate them to a cosmetic preparation containing liposomes.

p { margin-bottom: 0.25cm; line-height: 120%; }

Author Response

1.Page 4, line 170 Table 2. It is not clear to which liposome preparation belong the data of Table 2. The authors report under M&M three different compositions. Please clarify.

Answer: Firstly, we selected a basic composition (30 mg of phospholipids, 10 mg of Chol, and 10 mg of VD3) to compare two liposomes preparation methods (Ethanol injection, Film dispersion-homogenization) at “Vitamin D3 liposomes preparation” chapter. The liposomes preparation in Table 2 were prepared by ethanol injection and film dispersion-homogenization method (30 mg of phospholipids, 10 mg of Chol, and 10 mg of VD3).The liposomes preparation method by ethanol injection had a higher EE% and uniform size, the results were shown in Table 2. We choose the ethanol injection method to prepare VD3 liposomes in the subsequent experiments.

 We then optimized liposomes formulation by response surface using drug EE%as an evaluation indicator, liposomes were prepared by ethanol injection method in this part. The optimalresult was 24.55 mg of egg PC, 5.16 mg of Chol and 11.34 mg of VD3. So, there were three different compositions in M&M.

2. Page 5, line 179, Table 3. The amount of eggPC reported under column X1 should be 10, 25 and 40 mg not 0, -1 and 1.

Answer: We had changed the virtual value to a real value in Table 3.

3. Which is the rationale behind the fitting analysis performed in order to correlate the EE% with the different proportions of lipids?

Answer:Cholesterol is the main component of liposomes bilayer membrane material, and it is a good membrane fluidity regulator, able to improve EE% and liposomes stability. This might be explained by the fact that if the ratio was small, cholesterol content was relatively high, and the difficulty of membrane formation would be increased, and the formed film is relatively flexible. If the ratio was big, the membrane fluidity of the liposomes would be affected.[1]In some article of, author analyzed the influence of phospholipid/cholesterol on the EE% and size.[2,3]The Mass Ratio of phospholipid and cholesterol was a significant influencing factor on drug EE%.

In response surface experiments of most articles, the amount of phospholipid, cholesterol and the ratio of lipid to drug were usually used as interrelated factors to optimize drug EE%. So, differentproportions of lipids could influence EE%. In our manuscript, the mass ratio of egg PC to Cholesterol is also a significant influencing factor on drug EE%.[3,4,5]

[1] Jiao-Jiao Y , Frank Q , Jun-Ling T , et al. Preparation, Characterization, and Antioxidant Activity Evaluation of Liposomes Containing Water-Soluble Hydroxytyrosol from Olive[J]. Molecules, 2017, 22(6):870-.

[2]Zeng C , Jiang W , Tan M , et al. Optimization of the process variables of tilianin-loaded composite phospholipid liposomes based on response surface-central composite design and pharmacokinetic study[J]. European Journal of Pharmaceutical Sciences, 2016:S0928098716300379.

[3]Jiao-Jiao Y , Frank Q , Jun-Ling T , et al. Preparation, Characterization, and Antioxidant Activity Evaluation of Liposomes Containing Water-Soluble Hydroxytyrosol from Olive[J]. Molecules, 2017, 22(6):870-.

[4]Wu Z , Guan R , Lyu F , et al. Optimization of Preparation Conditions for Lysozyme Nanoliposomes Using Response Surface Methodology and Evaluation of Their Stability[J]. Molecules, 2015, 21(6):741.

[5]XiongY , Guo D , Wang L , et al. Development of nobiliside A loaded liposomal formulation using response surface methodology[J]. International Journal of Pharmaceutics, 2009, 371(1-2):197-203.

4.Why did the authors never prepare liposomes without cholesterol? Or in which the amount of VitD3 incorporated is equivalent to the amount of cholesterol reduced.

Answer:Cholesterol serves tocondense and order the polar lipids, thereby thickening,stiffening, and strengthening the bilayer and reducing its passivepermeability to small molecules even while increasing itsfluidity.[1]Cholesterol is an important component of liposomes bilayer membrane, it could improve the EE% and stability of liposomes.Therefore, we did not prepare liposomes without cholesterol.

 We optimized liposomes formulation by response surface using drug EE%as an evaluation indicator, liposomes were prepared by ethanol injection method in this part. The optimalresult was 24.55 mg of egg PC, 5.16 mg of Chol and 11.34 mg of VD3.The resultsindicated that a certain amount of cholesterol incorporated could improve the EE% of VD3.

[1] Lange Y ,Tabei S M A , Ye J , et al. Stability and Stoichiometry of Bilayer PhospholipidCholesterol Complexes: Relationship to Cellular Sterol Distribution and Homeostasis[J]. Biochemistry, 2013, 52(40):6950-6959.

5. Page 11, lines 286-287. Although the results are promising it is not correct to extrapolate them to a cosmetic preparation containing liposomes.

p { margin-bottom: 6.25px; line-height: 120%; }

Answer: I have corrected the incorrect statement. The manuscript format appears to have been reedited before being returned to me.

Reviewer 2 Report

Overall, interesting work.

Please expand on reference 14 in the introduction.

Was the transdermal absorption study using rat skin also covered under the IACUC? I'm assuming it was, but you should state this.

Table 5 is incredibly difficult to read. When stability was studied, did you measure particle size and PDI? Please add these values.

It is hard to see any differences in Figure 3, and you are making qualitative judgments using only your eye. I would try to quantify this somehow.

Author Response

We would like to express our sincere thanks to the reviewers for the constructive and positivecomments and suggestions. Your suggestion makes our manuscript more professional.

Major concern:

1. Please expand on reference 14 in the introduction.

Answer: Thank you for helping us improve the logic of the manuscript, I have expanded the introduction according to your suggestion.

Skin exposed to UV was the major access to obtain VD3 for most humans. But the VD3 photosynthesis, erythema and CPDs formation were all in the same spectra range of UVB, the photosynthesis of VD3 could not dissociated from photodamage.However, VD3 could be transformed under UV, it only occurs in the first few minutesupon exposure to UV, and longer exposures add nothing toVD3 stores despite increasing DNA damagein a linear fashion. So, naturally occurring VD3 under UV is insufficient to prevent photoaging. We need to supply additional VD3 for skin.

2. Was the transdermal absorption study using rat skin also covered under the IACUC? I'm assuming it was, but you should state this.

Answer: All animal experiments were approved by Institutional Animal Care and Use Committee. I have added it the transdermal absorption study.

3. Table 5 is incredibly difficult to read. When stability was studied, did you measure particle size and PDI? Please add these values.

Answer: Table 5 has been reedited for readability, and thechange of particle size and PDI of VD3liposomes were also supplied in Table 5.

4. It is hard to see any differences in Figure 3, and you are making qualitative judgments using only your eye. I would try to quantify this somehow.

Answer:We have quantitatively analyzed the treatment effect of photoaging by VD3 liposomes using the skin color and scale. The results and discussion were shown in manuscript.

Table 8. Assessment of photographs of photoaging rats.

Assessment

PBS

VD3 solution

VD3 liposomes

Aloe-gel

Male

Wrinkles a

Na

Na

Na

Na

Mottles a

Na

1

Na

1

Photodamage a

3

2

Na

Na

Roughness a

4

3

3

4

Overall   appearance b

+++++++

++++++

+++

+++++

Female

Wrinkles a

Na

Na

Na

Na

Mottles a

Na

1

Na

1

Photodamage a

1

Na

Na

Na

Roughness a

3

2

1

1

Overall   appearance b

++++

+++

+

++

a Based on a five-point scale:   1, excellent; 5, poor. Na: not available.

b + and ++, satisfactory; +++   and ++++, acceptable; from +++++ to +++++++, poor.

Figure 4.Color of rats’ skin after treatment. * p<0.05, VD3 liposomesVSVD3 solution for female rats; ## p<0.01, VD3 solution, VD3 liposomes or aloe-gelVS PBS for male and female rats.

Round 2

Reviewer 2 Report

Additions are sufficient. After language and ponctuation corrections, the article should be accepted.

Author Response

Review:

Additions are sufficient. After language and ponctuation corrections, the article should be accepted.

Answer: Thanks for your nice comments on our manuscript. These comments have contributed a lot to improve the quality of our article. According to your suggestions, we have made modifications to our manuscript. We feel really sorry for our carelessness. Manuscript has been checked by a native English speaking colleague.